# Kinematics of Cervical Spine during Rowing Ergometer at Different Stroke Rates in Young Rowers: A Pilot Study

**DOI:** 10.3390/ijerph19137690

**Published:** 2022-06-23

**Authors:** Valerio Giustino, Daniele Zangla, Giuseppe Messina, Simona Pajaujiene, Kaltrina Feka, Giuseppe Battaglia, Antonino Bianco, Antonio Palma, Antonino Patti

**Affiliations:** 1Sport and Exercise Sciences Research Unit, Department of Psychology, Educational Science and Human Movement, University of Palermo, 90144 Palermo, Italy; valerio.giustino@unipa.it (V.G.); daniele.zangla@unipa.it (D.Z.); kaltrina.feka@unipa.it (K.F.); giuseppe.battaglia@unipa.it (G.B.); antonino.bianco@unipa.it (A.B.); antonio.palma@unipa.it (A.P.); antonino.patti01@unipa.it (A.P.); 2Department of Coaching Science, Lithuanian Sports University, LT-44221 Kaunas, Lithuania; simona.pajaujiene@lsu.lt; 3Regional Sports School of CONI Sicilia, 90141 Palermo, Italy

**Keywords:** joint mobility, range of motion, biomechanics, kinematics, sport performance, cervical mobility, cervical range of motion, rowing, stroke cycle, stroke rate

## Abstract

Background: Research on biomechanics in rowing has mostly focused on the lumbar spine. However, injuries can also affect other body segments. Thus, the aim of this pilot study was to explore any potential variations in the kinematics of the cervical spine during two different stroke rates on the rowing ergometer in young rowers. Methods: Twelve young rowers of regional or national level were recruited for the study. The experimental protocol consisted of two separate test sessions (i.e., a sequence of 10 consecutive strokes for each test session) at different stroke rates (i.e., 20 and 30 strokes/min) on an indoor rowing ergometer. Kinematics of the cervical spine was assessed using an inertial sensor capable of measuring joint ROM (angle of flexion, angle of extension, total angle of flexion–extension). Results: Although there were no differences in the flexion and total flexion–extension movements between the test sessions, a significant increase in the extension movement was found at the highest stroke rate (*p* = 0.04, *d* = 0.66). Conclusion: Young rowers showed changes in cervical ROM according to stroke rate. The lower control of the head during the rowing stroke cycle can lead to a higher compensation resulting in an augmented effort, influencing sports performance, and increasing the risk of injury.

## 1. Introduction

Rowing is a strength-endurance sport characterized by a cyclical movement, i.e., the rowing stroke cycle, that although is apparently automatic, is actually a complex and structured movement that requires a high level of coordination, strength, power, and balance, as well as aerobic capacity and anaerobic capacity [1,2,3,4]. In both rowing disciplines, i.e., sweep rowing and sculling, four phases can be distinguished in the rowing stroke, such as catch, drive, finish, and recovery [5].

The technical skill of the rowing stroke was achieved and optimized with high levels of intensive training both in water and on land [6]. Regarding the land training, the rowing ergometer is the machine that allows replicating the rowing stroke, and it represents the specific training off water for rowers in order to improve the technique and the performance of this complex motor skill [1,7]. Several studies analyzed different biomechanical parameters (both kinematics and kinetics) of the stroke on a rowing ergometer, such as length and duration of the stroke and of its phases, power of the stroke, pulling forces on the handle, pushing forces on the foot stretcher, trajectory of movement of the handle, length of displacement of the seat, and joints angles [1,7].

Research into the kinematics of rowers shows many studies that detected differences in stroke rowing between athletes (e.g., junior vs. senior or elite vs. beginning) [1,8], between types of rowing ergometers (e.g., fixed head vs. moving head) [9], and between different stroke rates [10].

Focusing on the kinematics of rowing stroke can be useful in preventing the risk of injury and improving sports performance [7,11,12]. In particular, the scientific literature agrees that lumbar spine injuries represent the most frequent injury in rowers. For this reason, previous research groups investigated the kinematics of rowing stroke in this region [13,14,15,16,17]. In a seminal work by McGregor et al. (2004), the lumbar spinal motion during three different stroke ratings was examined, finding kinematics changes at higher rowing intensities [10]. However, other less common injuries can affect other anatomical sites such as upper and lower limbs, chest, and shoulders [5,18]. To the best of our knowledge, no studies have investigated the biomechanics of the upper spine during rowing ergometer. However, the biomechanics of this area during the rowing stroke cycle may need to be investigated further. Indeed, the kinematics of the cervical spine could be of interest not only with regard to injury prevention but also relevant to obtaining the best possible performance [19], avoiding compensatory movements that could negatively affect the health of the lumbar spine and also the competition.

Thus, based on previous biomechanical studies on the lumbar spine at different stroke ratings, this pilot study aimed to explore any potential variations in the kinematics of the cervical spine, in terms of cervical range of motion (ROM) (i.e., flexion and extension), during two different stroke rates on rowing ergometer in young rowers.

The choice to consider young rowers for our study is founded on the fact that previous research has shown differences in both kinematic and kinetic parameters in rowing technique between young and senior rowers and also on the basis of the higher rate of injuries that occurs in young rowers compared to seniors [1,8,20].

The hypothesis of our study is that young rowers could show higher cervical ROM at the highest stroke rate due to their lack of rowing experience.

## 2. Materials and Methods

### 2.1. Participants

Twelve young rowers (Male: 5; Female: 7; age: 13 ± 0.85 years; height: 156.42 ± 8.53 cm; weight: 51.37 ± 11.17 kg) from the CUS Palermo rowing team (Centro Universitario Sportivo di Palermo, Palermo, Italy) were recruited for the study. The Sport and Exercise Sciences Research Unit of the University of Palermo first presented the research to the rowing university sports team and, following the participation acceptance, it was presented to the athletes and their parents, requesting their participation on a voluntary basis after completing and signing informed consent. The inclusion criteria were the following: athletes of regional or national level; athletes belonged to categories under 15; athletes from 2 to 3 years of rowing experience; athletes regularly trained on the rowing ergometer. Participants were excluded in case of any kind of injury at the time of the study or injuries in the 6 months prior to enrollment in the study. Table 1 shows the characteristics of the participants.

As minors, the parents of the participants provided informed written consent to participate in the study. The study, conducted in accordance with the recommendations of the Declaration of Helsinki, has been approved by the Bioethics Committee of the University of Palermo (n. 59/2021).

### 2.2. Experimental Protocol

For this experimental protocol, we adopted the methodological approach of previous studies on rowing ergometers, assuming the bilateral symmetry of the rowing stroke movement and evaluating the kinematics in the sagittal plane [1,21].

As described in detail below, kinematics of the cervical spine was assessed on an indoor rowing ergometer (Concept2, Model C; Concept2 Inc., Morrisville, VT, USA) during two separate test sessions of different stroke rates using a wireless, computer-aided inertial sensor (Moover^TM^, Sensor Medica, Guidonia Montecelio, Rome, Italy). This device is capable of measuring joint ROM at a data acquisition frequency of 1 KHz with the related software (freeStep^®^, Sensor Medica, Guidonia Montecelio, Rome, Italy).

In particular, during the experimental protocol, each participant was asked to perform 5 min of warm-up and subsequently to perform the two test sessions with a rest period of 5 min between them. Each test session consisted of a sequence of 10 consecutive strokes at the stroke rate of 20 strokes/min (spm) for the first test session (T20) and 30 strokes/minute (spm) for the second test session (T30), respectively. The choice to adopt the stroke rates of 20 and 30 spm for the test sessions was based on the fact that these cadences are within the typical range of training and competition [9,22].

During each test session, the inertial sensor was positioned in the middle of the frontal bone of the skull and above the bridge of the nose and fastened around the head with an elastic band. The following cervical ROM parameters in the sagittal plane were recorded: angle of flexion (F), angle of extension (E), and total angle of flexion–extension (F–E). The same researcher carried out all the measurements. All the protocol was carried out under the supervision of a certified European rowing coach.

### 2.3. Statistical Analysis

The Shapiro–Wilk test was carried out to evaluate the distribution of data.

For statistical analysis, the average value of the 10 strokes of all parameters (i.e., F, E, F–E) was considered for each test session. Hence, the means and standard deviations of F, E, and F–E were calculated.

The paired sample *t*-test was used to examine differences in the parameters considered between the two test sessions. The *p*-value was set statistically significant at *p* < 0.05. The effect size was calculated through the Cohen’s *d* for paired sample using G*Power software 3.1.9.2 (Heinrich Heine University, Düsseldorf, Germany) considering the effect size conventions reported by Cohen (i.e., small effect: *d* ≥ 0.2; medium effect: *d* ≥ 0.5; large effect: *d* ≥ 0.8) [23].

Data were analyzed using IBM SPSS 23.0 (IBM Corporation, Armonk, NY, USA).

## 3. Results

The Shapiro–Wilk test revealed that data were normally distributed.

The paired sample *t*-test showed a significant difference in the E movement between T20 and T30 (*p* = 0.04, *d* = 0.66), as shown in Figure 1, whether no differences were found both in the F and F–E movements. Table 2 summarizes the values of the statistics.

## 4. Discussion

The aim of this pilot study was to explore any potential variations in the kinematics of the cervical spine in the sagittal plane by comparing movements of cervical flexion and extension during two separate tests on a rowing ergometer at 20 and 30 spm in young rowers. Our hypothesis was partially confirmed because young rowers showed a significant increase in the angle of cervical extension at the highest stroke rate, although no changes in the flexion movement were found.

It should be mentioned that the majority of studies on spine kinematics in rowers concern the lumbar region and, to the best of our knowledge, no research has investigated the upper spine, and this precludes the comparison of the results of this study.

Notwithstanding the foregoing, our findings can be explained by factors such as the performance level, the competitive level, and the rowing experience, as described below [10,11,24,25,26,27].

Regarding the performance level, a pivotal work on the kinematics of the lumbar spine in rowing reported greater spine changes at higher stroke ratings [10]. In detail, McGregor et al. (2004) investigated the lumbopelvic and lumbothoracic spinal motion at different stroke rates on an ergometer (i.e., 17–20, 24–28, and 28–36 spm, respectively) in a group of ten collegiate level rowers finding, at higher stroke rates, a significant reduction in the anterior lumbopelvic rotation at the catch phase, and, although not significant, an increase in the posterior lumbopelvic rotation [10]. Moreover, the authors detected a similar but not significant reduction in the lumbothoracic flexion at the catch phase and no changes in the angle of the lumbothoracic extension [10].

As for the competitive level, Hase et al. (2004) detected a significantly lower trunk extension and lumbar flexion in competitive rowers compared to non-experienced rowers [27]. Similarly, Smith and Spinks detected differences in biomechanical performance variables between novice, good athletes, and elite rowers, emphasizing the influence of rowing level on stroke quality [28].

Cerne et al. (2013) investigated the differences in kinematics and kinetics parameters by combining the two abovementioned factors, which are the performance level and the competitive level [1]. In particular, the authors analyzed rowing biomechanical parameters at different stroke rates (i.e., 20, 26, and 34 spm) between participants of different competitive levels (i.e., elite, junior, and non-rowers) and found that elite rowers use a similar technique at all stroke rates; this result was also comparable in junior rowers, while technique differences were found in non-rowers [1].

Lastly, among the factors that may explain the outcomes we found, it is worth mentioning the lack of rowing experience in young athletes that can influence rowing kinematics [11]. Indeed, although the participants recruited were at the regional or national level, they only had 2 or 3 years of experience, and for this reason, the young rowers may not have achieved a good command of the technique that the complex movement of rowing requires [1]. As a matter of fact, young rowers often show a higher annual incidence of injuries than seniors because of the lack of experience or the inadequate training, as reported in a recent review by Thornton et al. (2017), in which authors analyzed rowing-specific injury in detail [20].

Inadequate training can lead to the acquisition of an inappropriate technique, which could negatively affect the performance and augment the risk of injury [1,7,11]. In fact, the cyclical movement of the rowing stroke induces both repetitive loading and unloading of the spine, causing back injuries, especially in the lumbar region, the most common type of injury that occurs in rowers [5,7,8,15,20].

Notwithstanding the role of the upper spine [29,30], no studies have investigated the effect on the cervical region in rowers so far, not only for the risk of injury but also because it could influence the movements of the lumbar spine. As a matter of fact, a recent study emphasized the importance of multi-segmental spine kinematics analysis in rowing [31]. Indeed, and this can further explain our findings, the rowing stroke cycle consists of two macro-phases corresponding to the drive phase and the recovery phase, and particularly, during the drive phase, there is a distal to the proximal extension of all spine regions, which is essential for transferring the force from the lower limbs to the upper body [31]. In an opposite way, during the recovery phase is detected a proximal to distal flexion sequencing in order to achieve the optimal position for the subsequent drive phase [31]. With regard to the increase in the cervical extension movement that we found, we suggested an altered motor pattern that may occur simultaneously, one as a consequence of the other, as follows: (1) during the recovery phase, young athletes who have a lower technique may not reach the maximum possible flexion, (2) and this leads to compensatory movements, such as the excess in the cervical extension movement, during the next phase trying to generate a greater force due to an incomplete loading in the previous phase. As a matter of fact, the magnitude of the lower spine flexion increases with increasing rowing activity time [15].

Despite the lack of research on this topic, cervical spine injuries can often occur in other water sports activities [32]. For instance, forced hyperextension represents the cause of cervical injury in bodysurfers [33]. Although there are no detailed data on the rate of injuries in the cervical spine, it is widely recognized that the third most frequent injury in rowers, which is shoulder pain, can depend on neck muscle overuse, and improper use of these could indirectly emphasize the other [20]. In another study in high school rowers monitored during an entire academic year, the rate of neck injuries in boys was equal to that of the shoulders, while in girls, it was equivalent to that of the arms, elbows, and upper back [34]. The incidence of neck injuries was also investigated in adult rowers in a study by Carron et al. (2017) [35]. The authors reported that among the symptoms experienced by adult rowers following a single-handed transatlantic rowing race, 83% were traumatic injuries, including neck pain [35].

Since the rowing stroke requires the involvement of segmental coordination of the entire kinetic chain (i.e., sequential order of body segment movements) in order to transfer energy from the lower to the upper limbs with load/unload of the spine [5,24,36], the effect on the cervical spine should be properly investigated.

## 5. Conclusions

Based on our outcomes, we can summarize that young rowers showed changes in cervical ROM according to stroke rate. Specifically, cervical extension increases as the stroke rate increases. It should be noted that between the test sessions at 20 and 30 spm, a rest period of 5 min was given; therefore, the result found can only be linked to the biomechanical model adopted by the young rowers.

Based on our findings, the lower control of the head during the rowing stroke cycle can lead to higher compensation of the head, increasing the effort and influencing sports performance and the risk of injury.

Due to the importance of joint ROM both on performance and prevention of injuries [37,38], further research is needed to clarify the relationship between cervical movements and performance, as well as to examine the implications of cervical spine injuries, as already investigated in other sports [39,40,41]. In particular, it is widely recognized that, in young athletes, the prevention of injuries and the search for the optimal technique for obtaining the maximum possible performance are fundamental characteristics in sport [42,43,44].

## 6. Practical Implications

The preliminary results of this study reveal a practical implication for coaches. In fact, they suggest that, right from the beginning, it would be appropriate to pay attention to the biomechanics of the rowing stroke cycle in young athletes who approach this sport. Indeed, the acquisition of a technique without compensatory movements could favor the improvement of the skill in a more rapid and efficient way. Moreover, this approach prevents its later correction, which may be more strenuous and, among other things, exposes young athletes to a higher risk of injury.

## 7. Strength and Limitations

We acknowledge the following limitations of the study: the small sample size recruited and the study of the kinematics of the cervical spine limited to movements of a single plane and only for the upper spine without considering the other spine regions.

We would underline that this is a pilot study aimed to explore any potential variations in the kinematics of the cervical spine, in terms of cervical range of motion (ROM), at different stroke rates in young rowers. As it is the first step of the research, it is characterized by a small-sized sample, and it was conducted in order to allow any modification of the main study and to plan the best protocol. Hence, based on our results, we would suggest that for further studies on this topic, it should also be useful to record data relating to angular velocity and angular acceleration of the cervical spine.

However, beyond the limitations of the study, it should be highlighted that biomechanics analysis in the sagittal plane during the rowing stroke cycle is basal to study and improve movement strategy with the aim of obtaining the best performance and preventing the risk of injury in rowers [31,45]; no previous studies have ever been carried out to evaluate the kinematics of cervical spine in rowers, and this represents the major novelty of the present research.

## Figures and Tables

**Figure 1 ijerph-19-07690-f001:**
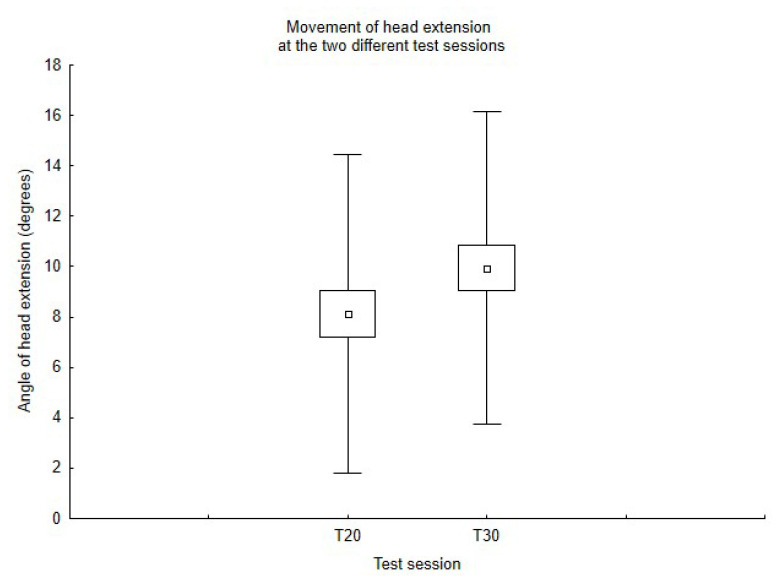
Movement of head extension at the two different test sessions.

**Table 1 ijerph-19-07690-t001:** Characteristics of the participants.

	*n*	Age	Height	Weight
M	5	13.00 ± 0.89	155.36 ± 8.09	50.13 ± 10.82
F	7	13.09 ± 0.83	156.09 ± 8.87	52.00 ± 11.49
TOT	12	13.00 ± 0.85	156.42 ± 8.53	51.37 ± 11.17

Legend. M, male; F, female; TOT, sample; *n*, number of participants.

**Table 2 ijerph-19-07690-t002:** Values of the cervical spine biomechanical parameters during the two test sessions.

	T20	T30	*p*	*d*
F (degrees)	7.01 ± 4.24	6.77 ± 4.65	n.s.	-
E (degrees)	8.12 ± 3.17	9.94 ± 3.10	0.04	0.66
F–E (degrees)	15.12 ± 3.97	16.70 ± 3.83	n.s.	-

Legend. F, cervical angle of flexion; E, cervical angle of extension; F–E, total angle of cervical flexion–extension; T20, test session at the stroke rate of 20 spm; T30, test session at the stroke rate of 30 spm.

## Data Availability

Data available on request.

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
