# Peer review of "Kinematics of Cervical Spine during Rowing Ergometer at Different Stroke Rates in Young Rowers: A Pilot Study"

_ijerph, 2022, doi:10.3390/ijerph19137690_

Round 1
Reviewer 1 Report
I think the results of this study are very helpful data for training young rowers and preventing injuries. I wrote down some comments and I hope they will help you in revising your manuscript.
- Introduction
The theoretical background of the study was adequately explained.
- Materials and Methods
Line 103-107: "In particular, during the experimental protocol each participant was asked to perform 5 minutes of warm-up and subsequently to perform the two test sessions with a rest period of 5 minutes between them. Each test session consisted of a sequence of 10 consecutive strokes at the stroke rates of 20 strokes/minute for the first test session (T20) and 30 strokes/minute for the second test session (T30), respectively."
Please explain why you chose T20 and T30 in this study. Or provide a reference.
- Discussion
In this study, the cervical angle of extension was significantly increased.
Please state the author's opinion whether the results of this study were because T20 and T30 were performed consecutively.
Author Response
I think the results of this study are very helpful data for training young rowers and preventing injuries. I wrote down some comments and I hope they will help you in revising your manuscript.
Dear Reviewer,
thank you for the time dedicated to review our manuscript. We thank you for your appreciation for our manuscript.
- Introduction
The theoretical background of the study was adequately explained.
We would like to thank the reviewer for the positive judgment.
- Materials and Methods
Line 103-107: "In particular, during the experimental protocol each participant was asked to perform 5 minutes of warm-up and subsequently to perform the two test sessions with a rest period of 5 minutes between them. Each test session consisted of a sequence of 10 consecutive strokes at the stroke rates of 20 strokes/minute for the first test session (T20) and 30 strokes/minute for the second test session (T30), respectively."
Please explain why you chose T20 and T30 in this study. Or provide a reference.
Dear Reviewer,
thank you for your positive comment which improves the quality of our manuscript.
The choice to adopt the stroke rates of 20 and 30 spm was based on the fact that these cadences are within the typical range of training and competition, as also reported in previous studies. Thanks to the reviewer's suggestion, we have reported this statement and the relative references in the new version of the manuscript (line 107-109).
- Discussion
In this study, the cervical angle of extension was significantly increased.
Please state the author's opinion whether the results of this study were because T20 and T30 were performed consecutively.
We thank the reviewer for the comment. As suggested, we reported our opinion on this result to the manuscript (line 220-223).
Reviewer 2 Report
The authors response well, so I have no more comment.
Author Response
The authors response well, so I have no more comment.
Dear Reviewer, we are delighted that our work has been appreciated by you.
Reviewer 3 Report
This is a resubmission of the work by the authors which measured sagittal parameters in young rowers at T20 and T30, showing more neck extension at 30 strokes per minute. While this is an interesting and novel pilot study our concerns remain the same, namely that cervical spine injury is very much a process of acceleration as much as it is about angles. The authors have very limited data here to draw any conclusions. Are they able at all to calculate any velocities or acceleration plots based on the saggital data?
Author Response
This is a resubmission of the work by the authors which measured sagittal parameters in young rowers at T20 and T30, showing more neck extension at 30 strokes per minute. While this is an interesting and novel pilot study our concerns remain the same, namely that cervical spine injury is very much a process of acceleration as much as it is about angles. The authors have very limited data here to draw any conclusions. Are they able at all to calculate any velocities or acceleration plots based on the sagittal data?
We agree with the reviewer, and we would thank the reviewer for the insightful comment. For this research, as it is a pilot study, we only have angular data, so we wanted to explore any differences in the angular kinematics of young rowers at different stroke rates. Therefore, we welcome the reviewer's comment and for the continuation of the research in addition to the angular data we will also record data relating velocity and angular acceleration. We added this issue as limitation of the study (line 246-253).
Round 2
Reviewer 3 Report
While it is worthwhile to note this as a limitation, velocities should be easily calculated and do not require additional testing. Unfortunately, whether this is a pilot study or not, there is not enough data here to make a meaningful addition to the literature.
Author Response
Dear Reviewer, since to the best of our knowledge there are no studies that have investigated angular kinematics of the cervical spine in rowers, the aim of this pilot study was to explore any kinematics changes, in terms of range of motion (RoM), in the cervical spine as the rate of strokes/minute varies. If the hypothesis of an increase in the (RoM) had been confirmed, that of associating these data with an acquisition of kinematic parameters relating to speed and acceleration was the second step.
This manuscript is a resubmission of an earlier submission. The following is a list of the peer review reports and author responses from that submission.
Round 1
Reviewer 1 Report
The manuscript presents a statistical analysis of data related to potential changes in the kinematics of the cervical spine.
The study group, even for a pilot study, is too small to achieve the assumed goal. The work did not provide any useful information for science.
There are no inclusion and exclusion criteria for research.
There are no clear presented practical implications of the study.
References - over 62% of references are older than 5 years and 45% of them are older than 10 years.
Author Response
The study group, even for a pilot study, is too small to achieve the assumed goal. The work did not provide any useful information for science.
Dear reviewer, in the new version of the manuscript you can find among the limitations of the study the size of the sample. However, we would like to specify that, as reported in the title, this is a pilot study. Research is still ongoing, and our goal is to achieve a consistent sample size.
There are no inclusion and exclusion criteria for research.
We apologize with the reviewer. We have clearly reported the inclusion and exclusion criteria in the methods section.
There are no clear presented practical implications of the study.
Thank you for your suggestion. At the end of the conclusion, we have added a paragraph with the practical implications of the study.
References - over 62% of references are older than 5 years and 45% of them are older than 10 years.
We thank the reviewer for the comment. The references reported refer to the seminal works on this topic. Furthermore, literature research on biomechanics applied to rowing has not provided us with recent works and, indeed, there would seem to be a lack of biomechanical analysis publications in this sport in recent years. Nevertheless, we did a further research and added some new references. However, iwe would be infinitely grateful if the reviewer has any works to suggest.

Reviewer 2 Report
This pilot study aimed to explore any potential variations in the kinematics of cervical spine during two different stroke rates on rowing ergometer in young rowers. They found that a significant increase in the extension movement was found at the highest stroke rate (p=0.04, d=0.66). Although this study is interesting and the manuscript is well-designed, their findings were limited. Therefore, I would suggest that the study could be accepted as a brief report, rather than full article.
1. Please shorten the introduction.
2. Please briefly describe the process of including study subjects3. Please add the table about the demographic feature of the study subjects.
4. Please revised the discussion, which should focus on your findings.
Author Response
We would like to thank the reviewer for defining our study interesting and the manuscript well-designed.
However, we are unable to change the type of this manuscript as a brief report rather than full article due to our ethics. In fact, the ethics of this research does not provide the publication of a brief report but primarily a pilot study which is this manuscript. We would like to clarify in fact that the research is still ongoing, and we reported in the title that it is a pilot study. In fact, we specify in the text that our results are preliminary.
Based on the above, however, we have responded to the other insightful comments as detailed below.
Please briefly describe the process of including study subjects.
We thank the reviewer for this suggestion. In the methods we described how the participant recruitment process took place.
Please add the table about the demographic feature of the study subjects.
Dear reviewer, thank you. We agree with the reviewer, and we add a table (Table 1) with the characteristics of the participants.
Please revised the discussion, which should focus on your findings.
We thank the reviewer for highlighting this point. In the new version of the manuscript with a more in-depth discussion of our findings.

Reviewer 3 Report
The authors assessed the rowing kinematics of 12 young rowers. The rowers underwent monitoring with an indoor rowing ergometer and inertial sensor. The sensor was placed on the frontal bone and recorded a number of head ROM parameters in the sagittal plane between two different types of stroke patterns. They found extension angles to be slightly different between the two speeds of rowing otherwise the total and flexion angles were similar.
Overall this is an interesting initial study. As the authors state, if this is the case, this is the first study to perform this on rowers. A much stronger analysis however would be needed to make conclusions of stroke rate or form predisposition to injury. As the authors mention in their limitations, these measurements were only in one plane. Even adding additional planes would not be strong enough. In future submissions or investigations we suggest the authors look at the RATE of change between flexion and extension as this is what has been shown to be deleterious in car accidents and sporting accidents. If there is some way to also measure muscle tension of the major neck flexors or extensors this would go a long way to make stronger conclusions.
Author Response
We thank the reviewer for evaluating our manuscript an interesting initial study.
A much stronger analysis however would be needed to make conclusions of stroke rate or form predisposition to injury. As the authors mention in their limitations, these measurements were only in one plane. Even adding additional planes would not be strong enough. In future submissions or investigations we suggest the authors look at the RATE of change between flexion and extension as this is what has been shown to be deleterious in car accidents and sporting accidents. If there is some way to also measure muscle tension of the major neck flexors or extensors this would go a long way to make stronger conclusions.
We thank the reviewer for highlighting this point with this insightful comment. We further emphasized that the limitation of the study is to have considered only the sagittal plane.
In the new version we have also reported that, on the basis of previous studies, the biomechanical analysis of the rowing cycle in the sagittal plane is baseline for the prevention of injuries and for the improvement of performance.
Furthermore, even though the reviewer suggested we perform this in future investigations, we wanted to follow the reviewer's suggestion because we think this will improve the quality of the manuscript and, already in this pilot study, we calculated the rate of change between sessions.
We would also like to thank the reviewer for the other suggestions for future research.

Reviewer 4 Report
Thank you for the opportunity to review this study. This study is considered necessary to prevent injuries and improve performance of rowers. I wrote the following comments on this study. I hope it will help you with your study.
1. introduction
- Please describe in detail why you chose young rower as the study subject.
2. Discussion
- Please describe in detail the reason why there was a significant difference between T20 and T30 in the angle of extension of the head of the study result.
- If there are past studies on upper spine biomechanics in adult rowers, please describe them in comparison with the results of this study.
- I think that it will be a strength of this study if objective data on neck injuries that occur frequently to young rower are described.
Author Response
We thank the reviewer for the comments that have improved the quality of the manuscript.
1. Introduction
Please describe in detail why you chose young rower as the study subject.
We thank the reviewer for highlighting this point. We have clearly reported why we recruited young rowers in our study.
1. Discussion
Please describe in detail the reason why there was a significant difference between T20 and T30 in the angle of extension of the head of the study result.
We thank the reviewer for this insightful comment. In the new version of the manuscript, we argued our findings with previous studies that support our explanation.
If there are past studies on upper spine biomechanics in adult rowers, please describe them in comparison with the results of this study.
Dear reviewer, to the best of our knowledge, there are no studies focusing on the biomechanics of the upper spine, not even in adults. In the new version of the manuscript we have also reported this statement in the discussion explaining that this limits the comparison with our results.
I think that it will be a strength of this study if objective data on neck injuries that occur frequently to young rower are described.
Dear reviewer, thank for your positive suggestion. Although there are no studies on the biomechanics of the cervical spine, in the new version of the discussion we have included data on the frequency of neck injuries in rowers.
